

# Artificial intelligence and cognitive diagnosis based teaching resource recommendation algorithm

Zhi Mao and Mingfang Li

Xi'an Technological University, Xi'an, China

## ABSTRACT

In the realm of advanced technology, deep learning capabilities are harnessed to analyze and predict novel data, once it has absorbed existing information. When applied to the sphere of education, this transformative technology becomes a catalyst for innovation and reform, leading to advancements in teaching modes, methodologies, and curricula. In light of these possibilities, the application of deep learning technology to teaching resource recommendations is explored in this article. Within the context of the study, a bespoke recommendation algorithm for teaching resources is devised, drawing upon the integration of deep learning and cognitive diagnosis (ADCF). This intricately constructed model consists of two core elements: the Multi-layer Perceptron (MLP) and the Generalized Matrix Factorization (GMF), operating cohesively through stages of linear representation and nonlinear learning of the interaction function. The empirical analysis reveals that the ADCF model achieves 0.626 and 0.339 in the hits ratio (HR) and the Normalized Discounted Cumulative Gain (NDCG) respectively due to the traditional model, signifying its potential to add significant value to the domain of teaching resource recommendations.

## INTRODUCTION

With the in-depth application and rapid development of information technology in the field of education, the rapid advancement of the strategy of "Internet plus education" and the rapid development of online teaching resources, interested teaching resources can be available through networked terminals anytime and anywhere. However, the massive teaching resources may cause confusion if users cannot quickly find the most suitable one among diverse teaching resources. At this time, the recommendation of teaching resources can better enable people to learn online and find more suitable materials for themselves. Combining deep learning with traditional recommendation algorithm can better discover users' needs. However, it is not enough to combine deep learning in the recommendation process. Compared with other resources, teaching resources will be more influenced by students' cognition (*Albus, Vogt & Seufert, 2021*). Moreover, with the change of students' cognition, teaching resources need to be constantly updated. Therefore, it is of great

Corresponding author
Zhi Mao, maozhi@xatu.edu.cn

significance to incorporate model of cognitive diagnosis into recommendation algorithm. Based on this, it is of great significance to design a teaching resource recommendation algorithm based on deep learning and cognitive diagnosis in detail, which can realizes personalized recommendation of teaching resources and improves users' experience and satisfaction.

The advancements in machine learning and artificial intelligence have led to significant developments in personalized recommendation systems. In the domain of education, the adoption of deep neural networks (DNN), Neural Collaborative Filtering (NeuCF), and cognitive diagnosis model (CDM) has shown promising results in the recommendation of teaching resources. DNN have proven to be a powerful tool in various applications, including recommender systems. They can effectively capture complex patterns in user behavior and item characteristics. For teaching resource recommendation, DNNs can utilize historical data on students' interactions with learning materials, such as textbooks, videos, or online courses, to suggest relevant resources tailored to individual learners' needs and preferences (*Joy & Pillai, 2022*; *Mubarak et al., 2022*). By employing techniques such as embedding layers and attention mechanisms, DNNs can map users and items into a latent space and provide personalized recommendations that enhance the learning experience. NeuCF is an extension of traditional collaborative filtering methods that leverages neural networks to model user-item interactions. Unlike matrix factorization techniques, NeuCF allows for non-linear interactions between users and items, enabling more accurate recommendations (*Li et al., 2022*; *Joshi et al., 2023*). In teaching resource recommendation, NeuCF can effectively capture the implicit feedback from students, such as clicks, views, or time spent on resources, to generate personalized suggestions. This method has shown improved performance compared to traditional collaborative filtering approaches, especially in scenarios with sparse data. CDM are a specialized class of educational data mining techniques that aim to infer students' cognitive skills or mastery levels for specific learning concepts (*Gao et al., 2022*; *Min & He, 2022*). In the context of teaching resource recommendation, CDM can identify students' strengths and weaknesses in various subjects and provide adaptive learning materials that align with their individual learning needs. By diagnosing students' cognitive abilities, CDM can recommend resources that address their knowledge gaps, fostering a more effective learning process.

DNNs offer high flexibility and can handle various types of data, including text, images, and sequential interactions, making them suitable for diverse teaching resource formats (*Safarov, Kutlimuratov & Abdusalomov, 2023*). NeuCF excels in collaborative filtering scenarios, where explicit user-item feedback is limited, as it can capture complex patterns and improve recommendation accuracy (*Linh, 2020*). CDM provides insights into students' cognitive mastery levels, enabling personalized resource recommendations that target specific knowledge areas (*Liang, De la Torre & Law, 2021*).

While DNNs and NeuCF rely on historical interaction data, CDM incorporates cognitive assessment data, allowing for a more comprehensive understanding of students' learning abilities. DNNs and NeuCF can scale well to large datasets due to their parallelizable nature and efficient training algorithms. CDM's applicability may be limited in environments

without extensive cognitive assessment capabilities or for subjects where cognitive skills are challenging to measure.

The application of the above model in teaching resource recommendation has shown tremendous potential in enhancing personalized learning experiences. DNNs excel in handling various data formats and interactions, while NeuCF captures complex patterns in collaborative filtering settings. CDM brings a new dimension by incorporating cognitive assessment data for more precise recommendations. The choice of method depends on the availability of data, the context of the recommendation system, and the desired level of personalization. Future research could focus on combining these methods or exploring hybrid approaches to leverage the strengths of each method for more effective and comprehensive teaching resource recommendation systems.

The main contributions of this article are:

(1) The generalized matrix GMF with element-wise product as interoperation and the MLP with neural network as interoperation are integrated, which makes the model have stronger feature combination and nonlinear ability.

(2) The NeuCF model is used in the recommendation of teaching resources, which can effectively learn the linear and nonlinear characteristics of courses and learning resources, and can be more effective for personalized recommendation of teaching resources.

(3) The proposed model integrates other auxiliary characteristic information of learners and course resources, introduces the difficulty factor of the course, and takes the relative sequence behavior of learning order into consideration for modeling.

In 'Design of recommendation algorithm in teaching resource based on deep learning and cognitive diagnosis', the design of the teaching resource recommendation algorithm is presented, encompassing the incorporation of DNN, NeuCF, and CDM. Moving on to 'Test in recommendation algorithm of teaching resource based on deep learning and cognitive diagnosis', comprehensive algorithm experiments are conducted. This section delves into the meticulous process of data collection and processing, shedding light on how the various algorithms are trained and subsequently compared in terms of their effectiveness. Finally, 'Conclusion' provides a succinct summary, highlighting the potential value brought forth by this research, while also addressing any identified shortcomings inherent in the study.

# DESIGN OF RECOMMENDATION ALGORITHM IN TEACHING RESOURCE BASED ON DEEP LEARNING AND COGNITIVE DIAGNOSIS

## Deep neural network model

Deep neural networks (DNN) are the most basic component of deep learning technology. They establish hidden layers, sequentially as well as the connection between the input layer and the hidden layer, and finally builds the connection between the hidden layer and the output layer, selects the activation function for each node of each hidden layer, and solves the weight of each connection and the offset value of each node. The essence of training is to solve it by combining back propagation and gradient descent. Given ($y1$,

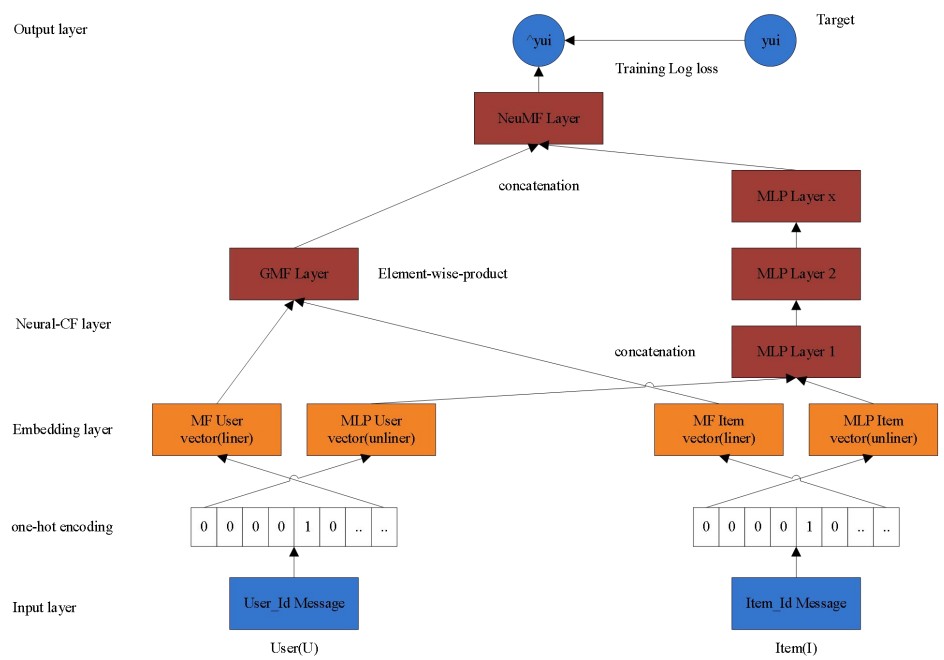

**Figure 1  NeuCF model.**

y2,., yn) and (x1, x2,., xn), to solve the deviation value of each connection weight and each neuron, it initializes each connection weight with a random number. Then, the predicted value calculated by neural network is compared with the true value of y. If the two values differ greatly, modify the connection weight of the current layer; If there is little difference between the two values, the weight of the lower layer will be modified. this step repeats until it is transferred to the weight value of first layer (*Liu et al., 2022b*).

## Recommendation model of neural collaborative filtering

The recommendation model of NeuCF consists of two parts: MLP and GMF. Among them, GMF learns linear features and MLP learns nonlinear features, so that NeuCF model has both linear and nonlinear modeling capabilities, and its generalization ability is enhanced (*Hao & Yang, 2022*). NeuCF recommendation model has the following two advantages: First, the ability of neural network is used to fit any function in theory, which can flexibly combine different features to increase or decrease the complexity of the model as needed; second, the NeuCF model integrates the generalized matrix GMF with element product as interoperability and MLP with neural network. This makes the model be capable of feature combination and nonlinear. Therefore, by applying the NeuCF model to teaching resource recommendation, it can effectively learn the linear and nonlinear characteristics of curriculum and learning resources, and make personalized recommendation of teaching resources more effectively. The NeuCF model is shown in Fig. 1.

The algorithm is mainly divided into two stages: the linear representation stage of interaction function and the nonlinear learning stage of interaction function. For a user u

and an item i in the database, and their characteristics (such as ID, user's gender, category, *etc.*). Since the ID is taken as the input in the CF model, the IDs of users and articles are used as the input in this method. Firstly, the feature of ID is encoded by one-hot, and the feature vector $\mathbf{v}_u^U, \mathbf{v}_i^I$ can be obtained. Then we can get the following four vectors after embedding, $\mathbf{p}_u^l, \mathbf{q}_i^l, \mathbf{p}_{u,}^r \mathbf{q}_i^r$, it should be pointed out that in the technical implementation details, different users and object embedding vectors share the same regularization coefficient.

$$
\begin{aligned}
\mathbf{p}_u^l &= f\left(\mathbf{v}_u^U\right) = \mathbf{P}_l^T \mathbf{v}_u^U \\
\mathbf{q}_i^l &= g\left(\mathbf{v}_i^I\right) = \mathbf{Q}_l^T \mathbf{v}_i^I \\
\mathbf{p}_u^r &= f\left(\mathbf{v}_u^U\right) = \mathbf{P}_r^T \mathbf{v}_u^U \\
\mathbf{q}_i^r &= g\left(\mathbf{v}_i^I\right) = \mathbf{Q}_r^T \mathbf{v}_i^I
\end{aligned}
\tag{2.1}
$$

Among them, f( )and g( )Represent the mapping functions of feature vectors respectively, $\mathbf{P}_l^T \in R^{M \times K}, \mathbf{Q}_l^T \in R^{N \times K}, \mathbf{P}_r^T \in R^{M \times K}, \mathbf{Q}_r^T \in R^{N \times K}$ respectively represent the entry vector of Hadamard product process of the weight matrix $(\mathbf{P}_l^T, \mathbf{Q}_l^T)$, the weight matrix of the input vector of linear feature extraction process $(\mathbf{P}_r^T, \mathbf{Q}_r^T)$. It is worth noting that this method can easily adjust the input without reducing the performance of the model. For example, the content features such as interests and hobbies existing in the system are used to represent users and items to solve the problem of cold start (*Xia & Hu, 2020*).

(1) Stage of linear representation in interaction function

The embedded layer above is a series of linear presentation layers. The left linear presentation layer $\mathbf{p}_u^l$ represents the implicit vector input by the user in the process of Hadamard product (LinearLayer-1), $\mathbf{q}_i^l$ indicates the input implicit vector of the item, then

$$
z_0^l = \phi_1^l\left(\mathbf{p}_u^l, \mathbf{q}_i^l\right) = \mathbf{p}_u^l \odot \mathbf{q}_i^l
\tag{2.2}
$$

Among them, $\phi_1^l$ represents the mapping function of Hadamard product process, C represents the element-by-element multiplication between vectors.

The right linear presents layer $\mathbf{p}_u^r$ is represented by implicit vectors representing users in the process of linear feature extraction (LinearLayer-r1~X), and $\mathbf{q}_i^r$ represents the implicit vector of the item, then

$$
\begin{aligned}
c_0 &= \begin{bmatrix} \mathbf{p}_u^r \\ \mathbf{q}_i^r \end{bmatrix} \\
c_1 = \phi_1^r(\mathbf{c}_0) &= a_1\left(\mathbf{W}_1^T \mathbf{c}_0 + b_1\right) \\
&\cdots\cdots \\
\mathbf{z}_0^r = \phi_X^r(\mathbf{c}_{X-1}) &= a_X\left(\mathbf{W}_X^T \mathbf{c}_{X-1} + b_X\right)
\end{aligned}
\tag{2.3}
$$

The whole process is realized by MLP structure of multilayer perceptron. Among them, $\mathbf{W}_X^T, b_X$ and $a_X$ respectively represent the hidden layer X of the extraction process in interactive function linear feature, the weight matrix, bias vector and activation function of the layer, The value range of X is [0,1], and the linear activation function is selected as activation function. In particular, when occasion X =1, the mapping dimension of the single linear presentation layer can be halved or reduced by 4 times in two ways, for example, from 64 → 16. MLP is used to implicitly model under this kind of cross

information. Theoretically, the explicit modeling is equally effective, such as the outer product operation in the ONCF model.

(2) Stage of nonlinear learning in interaction function

The nonlinear learning layer sends the complete linear representation of the interaction function into the deep neural network to carry out the process of nonlinear learning. Each layer can learn the high-order specific structure of the potential interaction between users and objects, while after learning, the complete user-object interaction function is obtained, which includes several non-linear layers 1-X in the middle, which is meant nonlinear layer -1, nonlinear layer -2…nonlinear layer -X respectively. It is also realized by MLP that the design pattern of hidden layer follows the principle of halving the dimension step by step. For single-layer nonlinear layer, two hierarchical dimension strategies provided in the process of linear feature extraction are applicable.

The size of the last layer in the hidden layer determines the ability of the model, which is called the predictive factor K. According to the predictive factors of different dimensions, MLP with different structures can be set, and generally MLP with 0~3 layers can be selected. For example, when K = 8, the hierarchical dimension structure of 3 layers of MLP is 32 $\rightarrow 16 \rightarrow 8$. This stage can be expressed as:

$$z_0 = \begin{bmatrix} z_0^r \\ z_0^l \end{bmatrix}$$
$$z_1 = \phi_1(z_0) = a_1\left(\mathbf{W}_1^T z_0 + b_1\right) \qquad (2.4)$$
$$\cdots\cdots$$
$$\phi_Y(z_{Y-1}) = a_Y\left(\mathbf{W}_Y^T z_{Y-1} + b_Y\right)$$

where $z_0$ and $z_Y$ represents the input vector in the nonlinear learning stage of the interaction function and the Y st nonlinear interactive function representation vector learned by the hidden layer, $\phi_1…\phi_Y$ represents the mapping function in this process. In addition, $W_1…W_Y, b_1…b_Y$ and $a_1…a_Y$ respectively represent the first layer of the hidden layer. L to the weight matrix of the Y st layer. The activation function selects the linear rectifier unit (ReLU), whose formula is as follows: $f(x) = \max(0, x)$. Because ReLU is unsaturated compared with other nonlinear activation functions, it has better biological adaptability, and more importantly, it is very friendly to sparse data, so that it can quickly improve performance when the model input is sparse data (*Liu et al., 2022a*).

The prediction layer performs scoring and generates a recommendation list. Firstly, it needs to solve the objective function of the optimization model. Because of the natural binary property of implicit data, the value of $y_{ui}$ can be put as the label -1 which represents the item U and the user i 0 is irrelevant, while the value of $\hat{y}_{ui}$ indicates the possibility of connection between the item u and the user i. Furthermore, the Top-N promotion based on implicit feedback can be regarded as a binary classification problem. The innovation lies in that it is different from the traditional square loss objective function, and the model can adopt the following loss function BCE similar to Log Loss. The adaptive learning algorithm called Adam is adopted to optimize the model. All the parameters of the model are learned in this process, and the negative sampling rate is 5 in the training process

**Table 1  Association matrix–Q matrix.**

| Resources | Knowledge | | | | | |
|---|---|---|---|---|---|---|
| | K1 | K2 | K3 | K4 | K5 | K6 |
| T1 | 1 | 0 | 0 | 0 | 1 | 0 |
| T2 | 0 | 1 | 1 | 0 | 0 | 1 |

(*Miao et al., 2022*).

$$L = -\sum_{\left(u,i \in Y + u\gamma^-_{\text{sampled}}\right)} y_{ui}\log\hat{y}_{ui} + (1 - y_{ui})\log(1 - \hat{y}_{ui}) \tag{2.5}$$

After the optimization of the model is completed, the score of final forecast is generated:

$$\hat{y}_{ui} = \sigma\left(\mathbf{h}^T \phi_Y(\mathbf{z}_{Y-1})\right) \tag{2.6}$$

Among them, $Y^+, Y^-_{sampled}$ respectively represent the trained positive sample set and the negative sample set partially sampled from the whole negative sample, L represents the optimized loss function in the process of model training, while $\mathbf{h}^T$ and $\sigma$ represent the weight matrix function and probability function of the model output layer. In this method $\sigma$ adopts sigmoid function, and its formula is:

$$\sigma(x) = \frac{1}{1 + \exp(-x)}. \tag{2.7}$$

## Cognitive diagnosis model

Cognition originates from modern psychometric theory and cognitive psychology. The purpose of diagnosis is to better understand the rules of individual psychological activities, diagnose and evaluate individual cognitive state, so that individuals can fully understand themselves and better adjust their cognitive state. The application of cognitive theory can be modeled directly from students' knowledge points, and then recommend teaching resources.

Q matrix is a matrix describing the relationship between items and attributes in the test, which is composed of the number of items and the number of attributes in 0-1 matrix. It is quoted into teaching resources, and its items are regarded as resources and attributes as knowledge points contained in resources, as shown in Table 1. Generally speaking, students' cognitive level is indirectly hidden, so it is impossible to calculate and measure them intuitively. Therefore, cognitive diagnosis is selected to estimate their cognitive level by combining Q matrix with students' actual learning situation, which ensures the interpretability of the results.

## Construction of teaching resource recommendation model

Based on the above, this article proposes a teaching resource recommendation model (ADCF) and applies it to recommendation of teaching resource. The ADCF model adds a

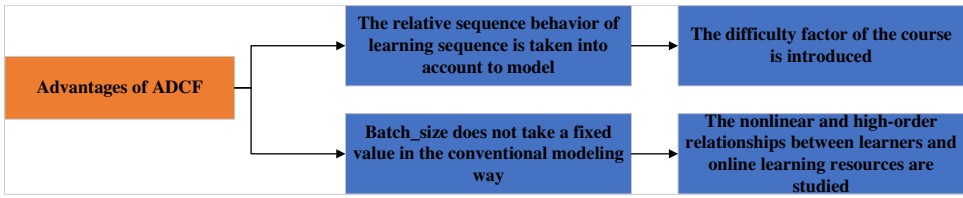

**Figure 2** The advantages of ADCF.

layer of cognitive diagnosis mechanism network between feature stitching layer and neural network layer. The weight of each dimension feature is obtained by Eq. (2.8) (*Li, Jing & Shi, 2022*).

$$
\begin{cases}
\text{Soft max}(Z_i) = \dfrac{e^{z_i}}{\sum_{i=1}^{k} e^{z_i}} \\
A_n = \text{Soft max}(X_n) \\
A_{\text{out}} = A_n \odot X_n
\end{cases}
\tag{2.8}
$$

In Eq. (2.8), $Z_i$ represents the ith input element of the Softmax function, while Soft*max* $(Z_i)$ represents the corresponding Softmax value of this element.

As shown in Fig. 2, the model proposed in this article makes the following improvements: first, it integrates other auxiliary characteristic information of learners and course resources, and introduces course difficulty factor; the second is to take into account the relative sequence behavior of learning sequence for modeling, that is, using the first N-1 effective interaction to predict the learner's NTH behavior; thirdly, in model training, because the interaction number of each learner is different, the batCH_size is not a fixed value in conventional modeling (such as 256 in NeucCF), but the interaction number of each learner is regarded as a BATch_size for dynamic learning. Fourth, the nonlinear and high-order relationship between learners and online learning resources is further studied. EGMF is not only used to model low-order linear interaction between courses, but also multi-layer perceptron is further used. Each hidden layer of multi-layer perceptron uses Relu nonlinear activation function. The hidden layer is set as Layers [64,32,16], and the number of neurons in each layer gradually decreases. The deeper hidden layer is the abstraction of the previous higher level, so as to learn the higher-order nonlinear relationship between the course and the learner. In addition, the kernel of GMF in the NeuCF model is the same as that of MF, because matrix decomposition into two matrices may produce negative numbers, which have no practical significance when used for recommendation and are not conducive to recommendation interpretation.

Therefore, in this article, the output of the feature stitching layer $X_n$ is regarded as the input of cognitive diagnosis mechanism, and the idea of cognitive diagnosis mechanism is used to distinguish the contribution degree of different historical interactive courses in predicting the target courses. The output of cognitive diagnosis mechanism $A_{\text{out}}$ is represented as the input of neural network. After repeated iterative training, the predicted

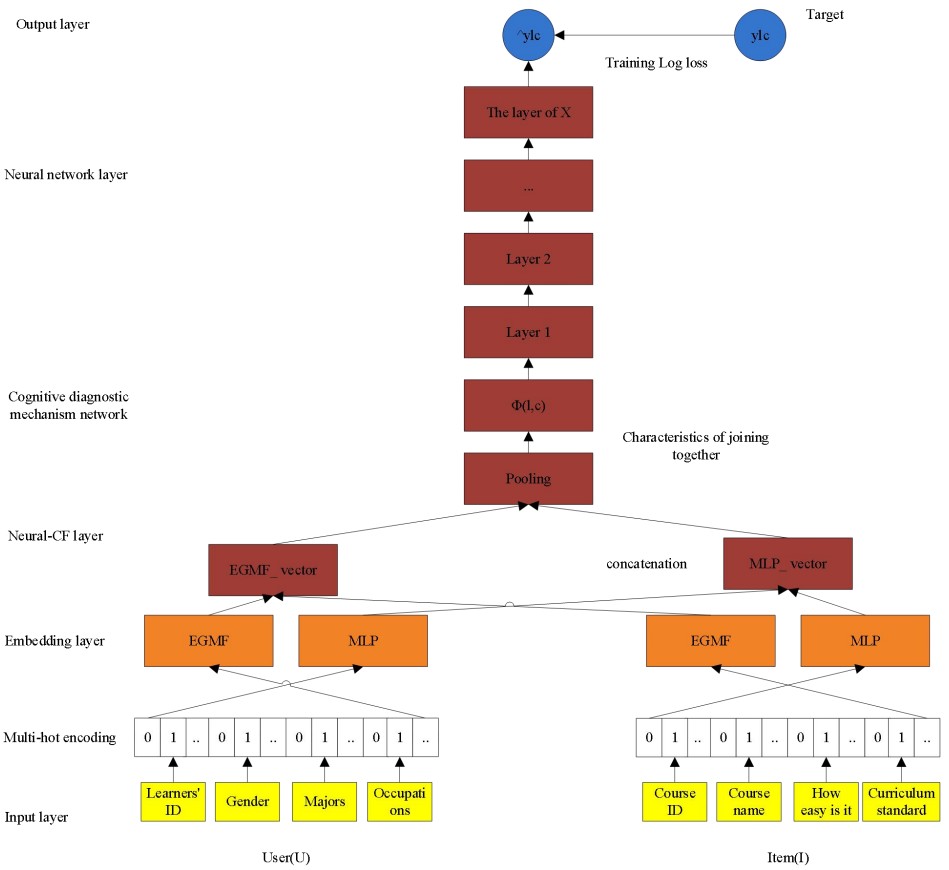

**Figure 3  ADCF model architecture.**

value obtained by an activation function is finally recommended. Figure 3 shows the architecture of ADCF model.

The mathematical model of ADCF is shown in Eq. (2.9).

$$
\begin{cases}
p_l^{EGMF} = \left(l_0^{EGMF} \oplus l_1^{EGMF} \oplus l_2^{EGMF} \oplus l_3^{EGMF}\right) \\
P_c^{EGMF} = \left(c_0^{EGMF} \oplus c_1^{EGMF} \oplus c_2^{EGMF} \oplus c_3^{EGMF}\right) \\
p_l^{MLP} = \left(l_0^{MLP} \oplus l_1^{MLP} \oplus l_2^{MLP} \oplus l_3^{MLP}\right) \\
q_c^{MLP} = \left(c_0^{MLP} \oplus c_1^{MLP} \oplus c_2^{MLP} \oplus c_3^{MLP}\right) \\
\phi^{EGMF} = p_l^{EGMF} \odot q_c^{EGMF} \\
\phi^{MLP} = \partial_L\left(W_L^T\left(\partial_{L-1}\left(\dots\partial_2\left(W_2^T\begin{bmatrix} p_l^{MLP} \\ q_c^{MLP} \end{bmatrix} + b_2\right)\dots\right)\right) + b_L\right) \\
A_{out} = \text{Softmax}\left(\begin{bmatrix} \phi^{EGMF} \\ \phi^{MLP} \end{bmatrix}\right) \odot \begin{bmatrix} \phi^{EGMF} \\ \phi^{MLP} \end{bmatrix} \\
\hat{y}_{lc} = \sigma\left(h^T A_{out} + b\right)
\end{cases}
\tag{2.9}
$$

In Eq. (2.9), $\oplus$ indicates connection, $l_i^{EGMF}$ and $l_i^{MLP}$ represent separately feature vectors of EGMF And MLP model learner, $l_0, l_1, l_2, l_3$ correspondence respectively learner$_{id}$, sex, profession, job $c_i^{EGMF}$ and $c_i^{MLP}$ represent EGMF and MLP feature vector of

course resource. $c_0, c_1, c_2, c_3$ represent {curriculum_id, name, complexity, label }. If EGMF and MLP share the embedded layer, the expressive ability of the model will be limited to a certain extent, so EGMF and MLP have their own embedded layers, among which $p_l^{EGMF} p_l^{MLP}$ and $q_c^{EGMF} q_c^{MLP}$ represent feature vectors of learner with auxiliary information and curriculum resource with auxiliary information for EGMF and MLP. $\phi^{MLP}$ and $\phi^{EGMF}$ indicate the potential relationships of characteristic respectively which EGMF and MLP have learned. In the part of cognitive mechanism, Softmax is used to calculate the different attention of each feature dimension, and then it is obtained by dot multiplication with corresponding features. With the input of the deep neural network ($A_{out}$), the predicted value is obtained through repeated iterative training. From the model, it can be seen that the auxiliary information of learners and courses runs through the whole model and jointly determines the performance of the model.

## TEST IN RECOMMENDATION ALGORITHM OF TEACHING RESOURCE BASED ON DEEP LEARNING AND COGNITIVE DIAGNOSIS

### Data preprocessing

#### *Web crawler*

(1) Development environment

The code environment is Python3.6, the compiler is PyCharm, and the dependent modules include: Selenuim, Bs4, re, Beautiful Soup, URLlib, Requests2.21.0, lxml and data persistence, *etc.* Here, Selenium+Web Driver is selected and WebDriver needs to correspond to the installed version of Chrome browser.

(2) Crawling process

In this article, massive open online course rich in learners' information, is selected, and the historical interactive data of learners' computer courses from 2014 to 2020 on MOOC is used as the experimental data set.

The implementation of the crawler is as follows: Firstly, determine the range of the ID of the crawling course, then use the starting value ID to splice the URL, such as ID = 100006, the spliced URL is https://www.imooc.com/u/100006/courses, afterwards, take the Requests third-party library to request the target website, and directly parse the html page to extract the data with lxml parsing library, which is of high efficiency. At the same time, in order to improve the efficiency of crawler, Multiprocessing is used for multi-process crawling. Moreover, the information that can be crawled from massive open online course's Internet includes not only the learners' own information such as gender, position, position, brief introduction and study duration, but also the learners' learning records of history courses, such as the courses they have studied, study progress, study notes, questions and answers, scores, comments, codes and other information. The process is shown in Fig. 4.

Crawler starts from the homepage, and then make further crawling from different categories. The specific process is as follows: First, obtain the URL of specific subjects under each category; Then, further crawl the detailed information of the course, including the course name, lecturer, course details, course review, course release time, course support

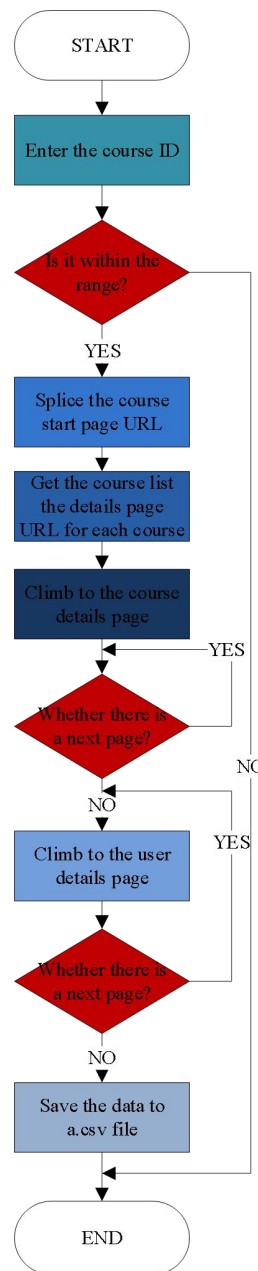

**Figure 4  Data crawl process.**

and course completion, and the learner ID of the review course; Finally, the obtained data is stored in the table of course_info.csv . The process is shown in Fig. 4.

Part of the data information of the crawled course details page includes the course ID, course name, course direction, course category, degree of difficulty, course hours, number of learners, comprehensive score of the course, number of questions and answers, number

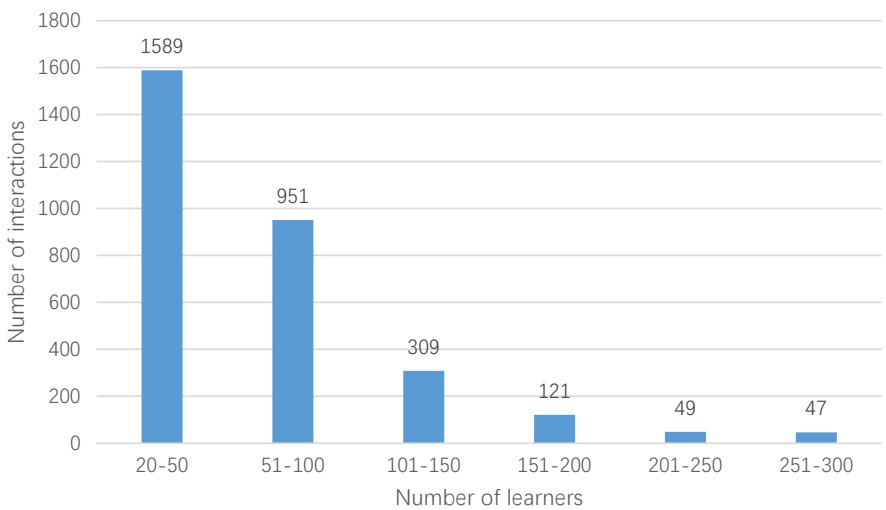

**Figure 5** Interaction number and number distribution.

of comments, number of notes, course chapters, number of code exercises, number of videos and number of evaluations, *etc.*, and is stored in the courseinfo.csv file.

The learner's learning history data is crawled including learner ID, nickname, gender, learning duration, learning experience, learning points, attention, fans, course ID, course name, interaction time, learning progress, learning notes, learning code, answer, location, position and other information, and they are stored in userinfo.csv file.

### Data processing

In this article, the crawler is used to crawl the relevant data from 2014 to 2020 in massive open online course, and the data of computer-related course is selected to construct the experimental data set. Records with more than 20 interactions are selected as the main object. Finally, the data set userlabelrl has 878 courses, 3,066 learners and 203,987 scoring data. There are 14 types of occupations: Java development, JS development, Linux system development, PHP development, Python engineer, UI designer, WEB front end, product manager, interaction designer, full stack engineer, page refactoring design, mobile development engineer, students and others. In addition, courses are divided into 10 categories: operating system IOS, UI design and multimedia, back-end development, front-end development, database, cutting-edge technology, mobile development, games, cloud computing and big data, maintenance and testing. Moreover, the difficulty of the course is divided into four levels: introductory, primary, intermediate and senior. Through statistics of script, the number of learners' interactions is segmented, and the number of learners in the corresponding interaction number range is counted separately. The details are shown in Fig. 5.

The data set from the perspective of data sparseness in Table 2 is analyzed below. Generally, sparsity of data set = 1 −number of effective interactions/(number of learners * number of courses). From this sparsity formula, the closer the value is to 0, the denser

**Table 2  Basic information of the UserLabelrL data set.**

| Data set | Number of course | Number of learners | Number of historical interaction records > 20 | Sparse degree |
|---|---|---|---|---|
| MOOC course resources | 878 | 3,066 | 203,987 | 92.42% |

the data set is, while the closer the value is to 1, the sparser it is. The sparsity of userlabelrl data set is about 92.42%, and the sparsity of MovieLens100M and pinterest data sets which are commonly recommended for research are 94.11% and 99.73%, respectively. Therefore, the userlabelrl dataset is relatively dense. Generally, the algorithm based on collaborative filtering has dense data set, which is more conducive to feature modeling, and can learn more hidden features from utility matrix, which is more conducive to recommendation (*Hill et al., 2023*).

## Design of test process
### *Selection of evaluation indicators*
Considering that recommendation of teaching resource is mainly used to present the most relevant curriculum resources to learners, that is, Top-k recommendation of curriculum list. To evaluate the list, the following two points need to be considered : First, put the results that learners are interested in at the top of the ranking, because learners are used to browsing from top to bottom; Second, the whole list is consistent with the learners' cognitive level. Therefore, HR and NDCG are selected as the evaluation indexes of the algorithm. HR intuitively measures whether the test items exist in the top k recommendation lists, while NDCG measures the gap between the predicted ordered list and the real interaction list. The results of the teaching resource recommendation in this article are only the top items, so HR and NDCG indicators are more suitable for evaluating the recommendation model proposed in this article.

The formula of HR index is shown in Eq. (3.1).

$$HitRatio@K@@ = \frac{NumberOfHits@K}{|GT|} \times 100\% \tag{3.1}$$

In Eq. (3.1), numerator represents the sum of the predicted number of test sets in each learner's Top-k list, denominator represents the number of test sets, and the ratio of numerator to denominator is the HitRatio (*Li, Cui & Jiang, 2022*).

The NDCG formula is shown in Eq. (3.2).

$$\begin{cases} CG_k = \sum_{i=1}^{k} reli \\ DCG_k = \sum_{i=1}^{k} \frac{2^{reli} - 1}{\log_2^{(i+1)}} \\ IDCG_k = \sum_{i=1}^{k} \frac{1}{\log_2^{1+i}} \\ NDCG_k = \frac{DCG_k}{IDCG_k} \end{cases} \tag{3.2}$$

In Eq. (3.2), i represents the position in the recommendation list, the value of k is from Top-K. In formula $CG_k$ (Commulative Gain), reli means the relevance of the recommendation results in position i. In this model, hit means reli = 1, miss means reli = 0. In formula $DCG_k$ (Discount Commulative Gain), reli = 1 represents the hit, so $DCG_k = \sum_{i=1}^{k} \frac{2^l - 1}{\log_2^{i+1}} = \sum_{i=1}^{k}$, reli =0 when missing, then $DCG_k = \sum_{i=1}^{k} \frac{2^0 - 1}{\log_2^{i+1}} = 0$. The location factor is introduced in $DCG_k$, which is more intuitive than $CG_k$ Evaluation .

$IDCG_k$ (Ideal Discount Commulative Gain), that is, all predictions hit, so reli =1 and the numerator is 1 constantly. Therefore, The formula $IDCG_k$ can be simplified as $IDCG_k = \sum_{i=1}^{k} \frac{1}{\log_2^{1+i}}$ $IDCG_k$ is the most idealistic $DCG_k$, therefore $DCG_k \in [0, IDCG_k]$.

$NDCG_k$ (Normalized Discount Communicative Gain), its essence is the ratio of $DCG_k$ and $IDCG_k$, the range is between [0,1], and the larger the value, the better the effect.

Generally, the recommendation of teaching resources will give a recommendation list, and reli represents the scores of the top k courses, the denominator is the offset of the position, there will be an attenuation coefficient in the further down position. While the higher the scores of the courses in the front, the higher the profits. The courses that exceed the cut-off value k have no effect on the results. Therefore, adding up the first k recommendation results is the cumulative discount gain called $DCG_k$, where Denominator $IDCG_k$ is the cumulative income under ideal conditions. In this test, after training the model with the training set, the performance of the model is tested with the verification set. Moreover, the standard is the coincidence degree between the prediction set given by the trained model and the real list in the verification set. The high coincidence degree between the prediction set and the verification set, The larger the value of $NDCG_k$, and the lower the coincidence degree, The smaller the value of $NDCG_k$.

### Cross entropy loss function

This algorithm uses the cross entropy loss function, and its principle is to quantify the accuracy of the classifier by punishing the wrong classification. The formula of cross entropy loss function is shown in Eq. (3.3).

$$L(Y, P(Y|X)) = -\log P(Y|X) = -\frac{1}{N} \sum_{i=1}^{N} \sum_{j=1}^{M} y_{ij} \log(p_{ij}) \tag{3.3}$$

In Eq. (3.3), $X$ is the input variable, $Y$ is the output variable, $L$ is the loss function, $N$ is the number of the sample size, $M$ is the number of possible categories. $y_{ij}$ is a binary indicator that represents the category. $j$ is an input instance about whether $x_i$ needs to input, $p_{ij}$ is the probability of input instances for models or classifiers $x_i$ Belong to category $j$. In addition, there are only two types of recommended results in this article. $\{0, 1\}$, the recommended value is 1, and the non-recommended value is 0, so the formula of cross entropy loss function can be simplified to Eq. (3.4).

$$L(\text{loss}) = -\frac{1}{N} \sum_{i=1}^{N} \left( y_i \log p_i + (1 - y_i) \log(1 - p_i) \right) \tag{3.4}$$

**Table 3  Experimental results of different algorithms.**

| Number | Algorithm | HR@10 | NDCG@10 | Loading data time | TIME |
|---|---|---|---|---|---|
| 1 | ADCF | 0.6262 | 0.3389 | 14.3 s | 12.9 s (training)+2.3 s (test) |
| 2 | IUNeu | 0.6084 | 0.3203 | 12.3 s | 11.4 s (training)+2.1 s (test) |
| 3 | NeuCF | 0.5975 | 0.3171 | 11.1 s | 11.2 s (training)+1.4 s (test) |
| 4 | ConvNCF | 0.5457 | 0.2918 | 16.9 s | 16.9 s (training)+2.6 s (test) |
| 5 | EGMF | 0.5923 | 0.3095 | 2.3 s | 8.1 s |
| 6 | GMF | 0.5841 | 0.3087 | 2.1 s | 7.3 s |
| 7 | MLP | 0.5812 | 0.3016 | 4.7 s | 11 s |

$y_i$ is the real category of input instance for $x_i$, $p_i$ is the probability that input instance $x_i$ falls into category 1. L(loss) means the average value of logarithmic loss of each sample, and the cross entropy loss function can be measured the similarity of $y_i$ and $p_i$.

## Analysis of experimental results

Experimental distributed framework (NVIDIA®) and parallel computing platform (CUDA®) are used, where GPU is used for accelerated training in hardware. The data sets are all userlabelrl data sets, the number of predictions k is set to 10, the learning rate is 0.001, and the number of iterations is 20 times. The ADCF proposed in this paper is compared with several algorithms such as NeuCF, IUNeu, ConvNCF, EGMF, GMF and MLP, and the results are shown in Table 3.

Table 3 reveals that ADCF not only takes into account the auxiliary information and the relative order of learning but also incorporates the cognitive diagnosis mechanism to discern the varying degrees of contribution of distinct interactive courses to the outcomes. It surpasses other models in HR and NDCG, which underscores that the amalgamation of the more nuanced cognitive diagnosis mechanism facilitates a more comprehensive discrimination of historical interaction records, thus fostering enhanced feature acquisition. This, in turn, substantiates the substantial advancements of the ADCF algorithm model over alternative recommendation systems, rendering it eminently suitable for personalized teaching resource recommendations.

## CONCLUSION

The exigencies posed by big data necessitate ingenious approaches in unearthing requisite patterns, given the copious volumes of data generated by each enterprise, which are emblematic of ultra-high dimensions. The intricacy and cognitive burden entailed in processing such prodigious data volumes are being ameliorated through the utilization of big data analysis and intelligent computing technologies. This article delves into the exploration of a recommendation algorithm for teaching resources, predicated on the convergence of deep learning and cognitive diagnosis. By synergistically amalgamating the deep neural network, the recommendation model of neural collaborative filtering, and the cognitive diagnosis model, a sophisticated recommendation model for teaching resources, grounded in the tenets of deep learning and cognitive diagnosis, is meticulously crafted. The

historical interactive data of computer courses on MOOC, spanning from 2014 to 2020, is procured as the experimental dataset using crawler technology. Subsequently, the ADCF algorithm, propounded herein, is appraised vis-à-vis NeuCF, IUNeu, ConvNCF, EGMF, GMF, and MLP. The findings evince that the performance of the ADCF algorithm surpasses that of its counterparts, and it demonstrates remarkable efficacy in the domain of teaching resource recommendation. Moreover, its deployment holds the promise of galvanizing the development of online learning platforms by endowing them with personalized resource recommendations, thus elevating user experience and satisfaction. Consequently, it bestows invaluable insights for the construction and advancement of online learning platforms.

## ACKNOWLEDGEMENTS

We thank the anonymous reviewers whose comments and suggestions helped to improve the manuscript.

### Funding
This study was supported by the New era Shaanxi University Red Culture Education System Innovation Research Base, key construction project of ''Great Ideological and Political Course'', project number DSZ01. The funders had no role in study design, data collection and analysis, decision to publish, or preparation of the manuscript.

### Grant Disclosures
The following grant information was disclosed by the authors:
The New era Shaanxi University Red Culture Education System Innovation Research Base, key construction project of ''Great Ideological and Political Course'', project number DSZ01.

### Competing Interests
The authors declare there are no competing interests.

### Author Contributions
- Zhi Mao conceived and designed the experiments, analyzed the data, prepared figures and/or tables, authored or reviewed drafts of the article, and approved the final draft.
- Mingfang Li performed the experiments, performed the computation work, authored or reviewed drafts of the article, and approved the final draft.

### Data Availability
The figures, tables, and code are available in the Supplementary Files.
The dataset is available at Zenodo: Ando, Edward. (2022). Doctoral School 2022: Data for 3D analysis lesson [Data set]. Zenodo. https://doi.org/10.5281/zenodo.7142405

## Supplemental Information

Supplemental information for this article can be found online at http://dx.doi.org/10.7717/peerj-cs.1594#supplemental-information.

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
