# Peer review of "Artificial intelligence and cognitive diagnosis based teaching resource recommendation algorithm"

_PeerJ Computer Science, doi:10.7717/peerj-cs.1594_

## Round 0.1 · original submission · Minor Revisions

Please take into account the reviewers' comments.

Reviewer 1 ·

Basic reporting

Refer to additional comments.

Experimental design

Refer to additional comments.

Validity of the findings

Refer to additional comments.

Additional comments

1. The abstract is too large. Needs to be precise.
2. A contribution subsection should be included after the introduction section to mention the researchers' contributions in bullet points.
3. The organization of the paper is missing.
4. A literature table or related work table must be included. There should be a clear mention of the limitations of the existing scheme and what the researchers are targeting to achieve.
5. The courses are selected from 2014-2020. If the data is available till 2020 then it is ok. Otherwise, needs to update.
6. The results in numbers in percentages should be included in the abstract of the proposed and existing schemes.
7. References should be updated. 2022 and 2023 references should be added.
8. The paper is very hard to read. The authors need to revise the paper and clearly mention their contributions (e.g., objectives, performance measures etc).

Reviewer 2 ·

Basic reporting

All the idea is present, but the narration is not well accomplished.
Introduction:
The subject isn't supported by literature in the field, and the main questions to answer are not clear and explicit.
The author should be acquainted with someone in the area who is proficient in English.

Experimental design

The major problems are the acronymous which are not defined.
Examples: MLP; and GMF.
Nevertheless, the description is present with no objection.

Validity of the findings

No comment.

Additional comments

No comment.

Reviewer 3 ·

Basic reporting

This paper presents a deep learning approach to aid in the recommendation of the massive available teaching resources.

The contributions of the work should be more clear. Also, the questions that you try to solve are not clear.

English is consistent, although some improvements can be made.

There's a lack of background and context, to place your work among other contributions. I advise a section, after 1.Introduction, to add these references, so the structure should be updated.

Experimental design

No comment

Validity of the findings

No comment

Additional comments

1) The abstract could be more concise to the aim of the study. Should mention some results of the study.

2) Many acronyms are not defined in text, such as HR and others.

3) References range until 2021. An update is crucial.

---

## Round 0.2 · accepted · Accept

The authors accurately revised the manuscript as reported by reviewers.

Reviewer 1 ·

Basic reporting

N/A

Experimental design

N/A

Validity of the findings

N/A

Additional comments

N/A

Reviewer 3 ·

Basic reporting

References have been updated as requested.
English writing is better as well.

Experimental design

no comment

Validity of the findings

no comment